# Current Use of Oral Anticoagulation Therapy in Elderly Patients with Atrial Fibrillation: Results from an Italian Multicenter Prospective Study—The ISNEP Study

**DOI:** 10.3390/jpm12091419

**Published:** 2022-08-31

**Authors:** Francesco De Stefano, Alberto Benassi, Alberto Maria Cappelletti, Francesco Donatelli, Damiano Regazzoli, Salvatore Tolaro, Francesca Perego, Angelo Silverio, Alessandra Scatteia, Pasquale Guarini, Santo Dellegrottaglie, Simona Mariani, Elpidio Pezzella, Gennaro Galasso, Francesco Caiazza

**Affiliations:** 1Division of Cardiology, Villa dei Fiori Hospital, 80011 Naples, Italy; 2Department of Cardiology, Hesperia Hospital, 41125 Modena, Italy; 3Department of Cardiology, Cardio-Thoracic-Vascular Department, IRCCS San Raffaele Scientific Institute, 20132 Milan, Italy; 4Department of Cardiac Surgery, Istituto Clinico S.Ambrogio, 20149 Milan, Italy; 5Cardio Center, IRCCS Humanitas Research Hospital, 20089 Milan, Italy; 6Division of Cardiology, Centro Cuore Morgagni, 95030 Catania, Italy; 7Division of Subacute Care, IRCCS Istituti Clinici Scientifici Maugeri, 20138 Milan, Italy; 8Department of Medicine, Surgery and Dentistry, University of Salerno, 84084 Salerno, Italy; 9Division of Cardiology, Aurelia Hospital, 00165 Rome, Italy; 10Heart Department, Pineta Grande Hospital, 81030 Caserta, Italy

**Keywords:** atrial fibrillation, elderly, non-vitamin k antagonist oral anticoagulants, vitamin K antagonists

## Abstract

Background: Atrial fibrillation (AF) is the most common heart arrhythmia, and its prevalence increases with age. Oral Anticoagulant Therapy (OAT) with non-vitamin K antagonist oral anticoagulants (NOACs) or vitamin K antagonists (VKAs) is essential to avoid thromboembolic events in AF. However, this treatment is associated with a high risk of bleeding and low adherence in elderly patients. Aim: The aim was to evaluate the real-world use of OAT in a population of patients aged ≥80 years in twenty-three Italian centers and to investigate the tolerance of and patient satisfaction with this therapy. Methods: The ISNEP Study is a multicenter cross-sectional study enrolling patients with AF and aged ≥80 years and treated with either NOACs or VKAs. A written questionnaire was administered to each patient to evaluate the adherence to and patient satisfaction with this therapy. Results: The study included 641 patients with a mean age of 85 (82–87) years. The use of NOACs was reported in 93.0% of cases, with the remaining 7.0% treated with VKAs. A history of stroke events was reported in five (11.1%) and one (0.2%) patients in the VKA and NOAC groups, respectively. The rate of referred ecchymosis/epistaxis was significantly higher in the VKA group compared to the NOAC group (*p* < 0.001). Patients receiving NOACs reported a substantial improvement in their quality of life compared to the VKA group. Conclusions: A small, but not negligible, proportion of elderly AF patients is still treated with VKAs. Patients treated with NOAC have a higher level of satisfaction with the therapy and complete adherence.

## 1. Introduction

Atrial fibrillation (AF) is the most common heart arrhythmia. Its prevalence increases with age, and about 70% of the affected patients are between 65 and 80 years old [1]. The prevalence of AF is also expected to increase in the next decades, given the prolonged life expectancy in Europe [2]. 

Since elderly patients have a significantly higher thromboembolic risk compared to the general AF population, advanced age accounts for two points in the CHA_2_DS_2_-VaSc (composite score of congestive heart failure, hypertension, age ≥75 years, diabetes mellitus, vascular disease, age 65–74 years, and sex category), which is the recommended and most commonly used score for predicting the individual thromboembolic risk [3]. 

Oral anticoagulation therapy (OAT) has become a cornerstone of the pharmacological treatment of AF patients in order to reduce the risk of stroke or systemic embolism [3]. In recent years, the use of vitamin K antagonists (VKAs) has been substantially reduced due to the availability of a novel drug class, non-vitamin K antagonists oral anticoagulants (NOACs), which demonstrated non-inferior efficacy compared to VKAs and a better safety profile with a lower incidence of both minor and major bleeding events [4,5,6,7]. 

The category of elderly is a very particular patient population characterized by a higher prevalence of co-existing comorbidities, frailty, cognitive impairment, and multiple therapies with several potential drug–drug interactions; each one of these components may lead to a labile risk–benefit ratio from OAT [8]. On the one hand, the thromboembolic risk during AF increases with age, but this evidence is counter-balanced by the limited therapeutic window of every anticoagulant drug, thus leading to the elevation of bleeding risk itself [9,10]. Indeed, a comprehensive evaluation of a single patient, including social, clinical, physiological, and pharmacological factors, is often needed in order to provide the best and safest choices to manage AF in patients with advanced age [11,12].

The safety and efficacy of NOACs in the elderly have not yet been clearly demonstrated due to the absence of dedicated randomized controlled trials (RCTs) selectively enrolling patients older than 75 years. Moreover, the inclusion of elderly patients in the registration RCTs of the four NOACs was limited, and the only available evidence derives from post hoc analyses [13,14,15,16,17].

Against this background, we aimed to report the real-world use of OAT for the prevention of thromboembolic events in an unselected multicenter cohort of patients with AF older than 80 years and to evaluate the adherence to therapy and the degree of satisfaction of this particular patient population.

## 2. Materials and Methods

### 2.1. Study Population

The ISNEP (Italian Study on NOAC use in Elderly Patients) was a multicenter cross-sectional study, promoted by the Italian Society of Accredited Hospital Cardiology (SICOA), aimed at evaluating the real-world use of OAT for thromboembolic prevention in patients with nonvalvular AF aged ≥80 years. Patients were enrolled in 23 participating centers throughout the Italian national territory from February 2019 to July 2021. Enrollment occurred at the beginning of an outpatient visit. In the cross-sectional analysis, all patients were evaluated with a medical interview and a survey containing questions about the assumptions of the therapy prescribed and the perceived effect on their quality of life. In addition, data were collected about bleeding events and ischemic/embolic events. 

The inclusion criteria were patients with nonvalvular AF, either paroxysmal, persistent, or permanent, with age ≥80 years, under OAT treatment (both NOACs or VKAs) starting at least one month prior to the study. The exclusion criteria were valvular AF, recent cardiac surgery or transcatheter AF ablation (within 1 month), and any pathological conditions limiting life expectancy to less than 6 months. 

This study was conducted according to the Declaration of Helsinki. All patients were informed of the nature and aims of the study and asked to sign an informed consent for the anonymous management of their individual data. Local Institutional Review Boards approved the study protocol according to the current Italian rules.

### 2.2. Data Measures

The patient baseline characteristics, including data on home pharmacological therapy, were collected at the time of recruitment. Baseline demographic, clinical, echocardiographic, and laboratory data were prospectively collected and recorded on an electronic case report form. Beyond age and sex, the following demographic and clinical features were reported: height, weight, body mass index (BMI), systolic arterial pressure (SAP), diastolic artery pressure (DAP), estimated glomerular filtration rate (eGFR) with the Cockroft–Gault formula [18], smoking status, the pattern of AF (paroxysmal, permanent), absence/presence of diabetes mellitus, dyslipidemia, arterial hypertension, peripheral artery disease (PAD), coronary artery disease (CAD), heart failure (HF), liver disease, and co-existing drug therapies. Echocardiographic examinations allowed us to stratify patients depending on left ventricular ejection fraction (LVEF) values of ≥50%, 40–49%, and <40%.

OAT data were collected for each patient, including the type of therapy (NOACs or VKAs) and NOAC molecule (apixaban, dabigatran, rivaroxaban, or edoxaban) with its own dosage (full or reduced, depending on the single molecule’s criteria for reduction).

Each patient was administered a questionnaire evaluating the adherence to therapy, the events occurring from the beginning of OAT, and the level of satisfaction with the therapy. In detail, the questions of the survey comprised: events referred for ecchymoses, epistaxis, and other forms of bleeding since the prescription of OAT; the impact of therapy on patients’ quality of life; and the levels of improvement (no impact, low, enough, complete). The survey also had questions aimed at defining the adherence of the patients to the therapy (periods of interruption, number of pills forgotten).

### 2.3. Statistical Analysis

The normal distribution of continuous parameters was tested with the Kolmogorov–Smirnov test and histogram assessment. Normally distributed variables were expressed as mean ± standard deviation and compared using the Student’s *t*-test; variables with a skewed distribution were reported as median and interquartile range and were compared with the Mann–Whitney U test. Categorical variables were reported as numbers and percentages and compared using the χ^2^ test, or Fisher’s exact test, when appropriate. 

For all tests, a *p*-value <0.05 was considered statistically significant. Analysis was performed using SPSS software version 25.0 (SPSS Inc., Chicago, IL, USA) and R version 3.5.1 (R Foundation for Statistical Computing, Vienna, Austria).

## 3. Results

The baseline characteristics of the study population are summarized in Table 1. The study included 641 patients, with a mean age of 85 (82–87) years. There were 275 (42.9%) male patients. Permanent AF was the most frequent, accounting for 78.6% of the overall population. 

The use of NOACs was reported in most of the cases (93.0%), whereas VKAs were reported in 45 (7.0%). The study population showed a high prevalence of cardiovascular risk factors, including dyslipidemia (40.2%) and diabetes (31.2%), with no difference between patients treated with VKAs or NOACs. 

The estimated glomerular filtration rate was 50 (42–66) mL/min and, as expected, was significantly higher in the NOAC group compared to the VKA group (*p* = 0.003). 

Coronary artery disease (CAD) and heart failure (HF) were reported in almost one-third of the overall population (30.9% and 28.2%, respectively). 

As expected, the median CHA_2_DS_2_-VASc score was 4, accounting for at least one-third of the population, without statistical differences between the two groups (Table 2). The distribution of the CHA_2_DS_2_-VASc in the overall population is depicted in Figure 1. 

The hemorrhagic risk was more heterogeneous, with a HAS-BLED score of <3 in 53% of cases and >3 in the remaining 47%.

The median length of the OAT treatment was 36 months and was longer, albeit not statistically significant, in the VKA group than in the NOAC group. The history of adverse clinical events is summarized in Table 2. A history of stroke was reported in five (11.1%) and one (0.2%) patients in the VKA and NOAC groups, respectively; transient ischemic attacks (TIA) were reported in five patients of the VKA group (11.1%) **vs.** eleven cases in the NOAC group (1.8%); finally, acute coronary syndromes occurred in four cases in the VKA group (8.9%) vs. thirteen cases in the NOAC group, with *p* = 0.007. No statistically significant differences were found between the two groups for major and minor bleeding events, including gastrointestinal (GI) bleeding events. In the NOAC group, no differences were found for each molecule.

The way the patients answered the survey about tolerance and quality of life during OAT statistically differed between the two groups. In fact, the referred ecchymosis/epistaxis events were more frequent in the VKA group (*p* < 0.001). The same was reported for the burden of the OAT in changing the quality of life, once again more pronounced in patients in VKA therapy (Table 3). The major degrees in the improvement of quality of life (the drug taken improved it enough or completely) were expressed more frequently in the NOAC group.

## 4. Discussion

After the introduction to the clinical practice of the NOAC drug class, the number of AF patients treated progressively increased, and it is still growing. These drugs act in a different way than VKAs because they directly inhibit the activated thrombin (dabigatran) or the activated X factor (apixaban, edoxaban, and rivaroxaban). All the registration trials showed that NOACs were non-inferior to warfarin in preventing thromboembolic events in AF patients, with a better safety profile. 

In this scenario, real-world prospective studies may provide valuable information about the patient characteristics, the prescription pattern, and the adherence and satisfaction of patients to therapy, with the aim to identify gaps within guidelines and implementation of current recommendations in daily practice.

The main findings of this cross-sectional study can be summarized as follows: -In elderly patients, there was still a significant portion of this population treated with VKAs;-The patients’ satisfaction with therapy was higher in the NOAC group.

The patients were being treated with NOACs for over 93% of the cases; the remaining patients were on VKAs. The baseline characteristics did not differ between the two groups, and most of the cases presented at least one comorbidity, such as CKD, peripheral vascular disease (PAD), dyslipidemia, diabetes mellitus, or systemic hypertension. Almost one-third of the population had a diagnosis of CAD and/or HF. 

It is important to underline how, in this picture, almost 1 out of 10 patients in our population were still in therapy with VKAs. This proportion was consistent with other important and wider registries evaluating OAT in elderly and very elderly patients in Japan and France [19,20]. There are possible explications for this evidence: first, some physicians prescribing VKAs may be led by therapeutic inertia: partly because patients were on VKA therapy already before the introduction of NOACs, and partly due to the physicians’ belief that in patients with advanced age, the management of anticoagulant therapy can be safer with more frequent monitoring of the International Normalized Ratio’s (INR) values. Both reasons could explicate the confirmation of a pre-existent VKA prescription. The problem is still current, as different hospital institutions have proposed a systematic review of the possible candidates to switch to NOACs from VKAs [21].

However, as recently reported by large metanalyses [22,23], the use of NOACs in these settings of patients is more effective than VKAs. The safety profile is as good as in younger patients, with the only increased risk being for nonfatal gastrointestinal bleeding [23]. It is important to note that, in geriatric populations, the time in the therapeutic range (TTR) of VKAs tends to be lower than in the general population: this means that most patients are exposed to an increased risk of thromboembolic and hemorrhagic events [24]. Additionally, VKA prescription can be motivated by the concern for the renal dysfunction typically present in elderly patients: in fact, even in our population, CKD was one of the most frequent comorbidities, and the hepatic metabolism of VKAs could lead the physicians to prefer drugs potentially less deleterious for these patients. However, growing evidence in the literature confirms that NOACs are equally safe and effective even in the context of CKD, especially when the criteria for the reduction in the single drug dosage are respected [25].

Although our design did not contemplate a prospective observation, the elderly patients in the population study were on OAT for a median of 36 months. At the time of the enrollment, data were collected about hospitalizations and/or diagnoses for ischemia events since the start of the therapy. From this point of view, the NOAC group showed a very low prevalence of ischemic events, such as stroke, TIA, and acute coronary syndromes, confirming the efficacy of this drug class even in patients aged > 80. A previous single-center trial, the REGINA Registry, already confirmed the efficacy of NOACs in this setting of patients [26]. 

Therefore, these results indicate how the concern about safety might lead to inappropriate use of VKAs in elderly patients: first, because the patients on VKA more often do not stay within the correct TTR, and, in turn, they are exposed to an unjustified risk of both ischemic and hemorrhagic events. 

Finally, most of the patients treated with NOACs expressed, in the dedicated questionnaire, the acknowledgment that anticoagulant therapy improved their quality of life, and their satisfaction with the NOAC therapy was high. This can be related to a more complete adherence to chronic therapy. Adherence to therapy is still a significant issue in patients treated with OAT [27], and the fixed-dose regimen of NOACs, without the need to modify the daily dose, together with the more tolerated side effects and fewer interactions than VKAs, are all promising elements to reach the most complete effects of the therapy.

Our study had some limitations: first, as mentioned before, the cross-sectional observational design, as well the limited sample size, do not allow us to compare the results obtained between the two OAT subgroups directly, but this was not the main objective of the study. In fact, our purpose was to see the current state of the prescription of OAT in the Italian national territory. Secondly, we acknowledge that the ischemic events report was conducted in a retrospective manner, and so the information retrieved could be affected by a recall bias. However, the study population is currently followed in a perspective analysis, which is ongoing, and this could allow us to have a more reliable interpretation of the future events and, of course, even compliance with the therapy for a longer period of observation. Moreover, we avoided multivariate analyses because of the limited statistical power and qualitative nature of the questionnaire administered to patients.

## 5. Conclusions

The ISNEP study provides a real-life overview of the current prescription pattern of NOACs and VKAs in a high-risk population of elderly AF patients in Italy.

A small but nonnegligible proportion of elderly AF patients is still treated with VKAs, probably due to clinicians’ therapeutic inertia, the reluctance to prescribe NOACs in patients with poor renal function, or the lack of INR monitoring with NOAC. Finally, the patients treated with NOACs showed a higher level of satisfaction with the therapy, which can be associated with complete adherence throughout the course of it.

## Figures and Tables

**Figure 1 jpm-12-01419-f001:**
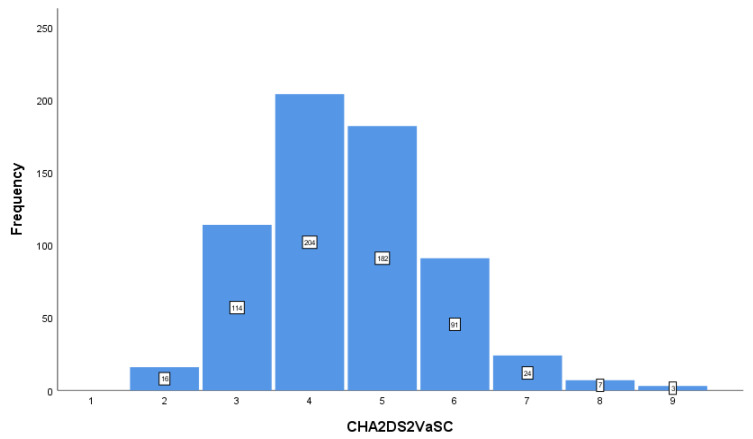
CHA_2_DS_2_-VASc score values in the overall population.

**Table 1 jpm-12-01419-t001:** Baseline characteristics of the study population.

	Overall Population (N = 641)	NOAC (N = 596)	VKA (N = 45)	*p*
Age, years	85 (82–87)	85 (82–87)	86 (82–88)	0.384
Male sex, N (%)	275 (42.9)	259 (43.5)	16 (35.6)	0.302
Weight, kg	71.0 ± 12.4	71.1 ± 12.4	68.5 ± 12.2	0.167
Height, cm	164.7 ± 77.1	164.8 ± 8.7	164.0 ± 10.0	0.557
BMI, kg/m^2^	25.5 ± 3.7	25.5 ± 3.7	24.8 ± 3.2	0.209
SAP, mmHg	140 (130–154)	140 (130–154)	135 (125–155)	0.493
DAP, mmHg	80 (75–90)	85 (75–90)	80 (70–80)	<0.001
Diabetes, N (%)	200 (31.2)	191 (32.0)	9 (20.0)	0.093
Dyslipidemia, N (%)	258 (40.2)	239 (40.1)	19 (42.2)	0.780
Smoking status, N (%)				
Never	447 (69.7)	412 (69.1)	35 (77.8)	0.049
Previous	123 (19.2)	113 (19.0)	10 (22.2)	
Active	71 (11.1)	71 (11.9)	0	
Paroxysmal AF, N (%)	96 (15.0)	93 (15.6)	3 (6.7)	0.268
Persistent AF, N (%)	41 (6.4)	38 (6.4)	3 (6.7)	
Permanent AF, N (%)	504 (78.6)	465 (78.0)	39 (86.7)	
CAD, N (%)	181 (28.2)	171 (28.7)	10 (22.2)	0.353
HF, N (%)	198 (30.9)	188 (31.5)	10 (22.2)	0.192
LVEF, N (%)				
≥50%	468 (73.0)	439 (73.7)	29 (64.4)	0.406
40–49%	140 (21.8)	127 (21.3)	13 (28.9)	
<40%	33 (5.1)	30 (5.0)	3 (6.7)	
Prior stroke/TIA, N (%)	91 (14.2)	77 (12.9)	14 (31.1)	<0.001
PAD, N (%)	123 (19.2)	115 (19.3)	8 (17.8)	0.803
eGFR, mL/min	50.0 (41.5–66.0)	51 (42–67)	45 (39–55)	0.003
Liver disease, N (%)	7 (1.1)	6 (1.0)	1 (2.2)	0.449
Concomitant medications				
Non-dihydropyridine CCB, N (%)	58 (9.0)	54 (9.1)	4 (8.9)	0.969
Antiarrhythmic drugs, N (%)	101 (15.8)	90 (15.1)	11 (24.4)	0.097
ACEi, N (%)	280 (43.7)	265 (44.5)	15 (33.3)	0.147
Beta-blockers, N (%)	481 (75.0)	451 (75.7)	30 (66.7)	0.178
Sartans, N (%)	156 (24.3)	140 (23.5)	16 (35.6)	0.069
Ranolazine, N (%)	16 (2.5)	16 (2.7)	0	0.266
Statins, N (%)	312 (48.7)	296 (49.7)	16 (35.6)	0.068
Prior bleeding, N (%)	31 (4.8)	28 (4.7)	3 (6.7)	0.553
Chronic use of antiplatelet agents or NSAIDs, N (%)	45 (7.0)	39 (6.5)	6 (13.3)	0.086

ACEi—Angiotensin Converting Enzyme Inhibitors, AF—Atrial Fibrillation, BMI—Body Mass Index, CCB—Calcium-Channel Blockers, CAD—Chronic Artery Disease, DAP—Diastolic Artery Pressure, eGFR—estimated Glomerular Filtration Rate, HF—Heart Failure, LVEF—Left Ventricular Ejection Function, NSAIDs—Non-Steroidal Anti Inflammatory Drugs, PAD—Peripheral Artery Disease, SAP—Systolic Artery Pressure, TIA—Transient Ischemic Attack.

**Table 2 jpm-12-01419-t002:** Adverse events during Oral Anticoagulant Therapy (OAT).

	Overall Population (N = 641)	NOAC (N = 596)	VKA (N = 45)	*p*
Duration of OAT since initiation, months	36 (17–45)	36 (17–42)	36 (13–81)	0.189
History of anemia during OAT, N (%)	36 (5.6)	34 (5.7)	2 (4.4)	0.723
History of GI bleeding during OAT, N (%)	16 (2.5)	14 (2.3)	2 (4.4)	0.385
History of major bleeding during OAT, N (%)	13 (2.0)	12 (2.0)	1 (2.2)	0.924
History of minor bleeding during OAT, N (%)	43 (6.7)	38 (6.4)	5 (11.1)	0.221
History of severe thrombocytopenia during OAT, N (%)	2 (0.3)	2 (0.3)	0	0.697
History of stroke during OAT, N (%)	6 (0.9)	1 (0.2)	5 (11.1)	<0.001
History of TIA during OAT, N (%)	16 (2.5)	11 (1.8)	5 (11.1)	<0.001
History of ACS during OAT, N (%)	17 (2.7)	13 (2.2)	4 (8.9)	0.007

ACS—Acute Coronary Syndrome, GI—Gastrointestinal, TIA—Transient Ischemic Attack.

**Table 3 jpm-12-01419-t003:** Patient questionnaire.

	Overall Population (N = 641)	NOAC (N = 596)	VKA (N = 45)	*p*
The patient is informed about drug–drug and food–drug interactions	529 (82.5)	492 (82.6)	37 (82.2)	0.189
The patient reports significant side effects from OAT	20 (3.1)	16 (2.7)	4 (8.9)	0.021
The patient reports ecchymosis	68 (10.6)	55 (9.2)	13 (28.9)	<0.001
The patient reports epistaxis	17 (2.7)	12 (2.0)	5 (11.1)	<0.001
The patient reports bleeding	22 (3.4)	20 (3.4)	2 (4.4)	0.699
OAT initiation improved the patient’s quality of life				
No	72 (11.2)	62 (10.4)	10 (22.2)	<0.001
Low	72 (11.2)	60 (10.1)	12 (26.7)	
Enough	276 (43.1)	260 (43.6)	16 (35.6)	
Complete	221 (34.5)	214 (35.9)	7 (15.6)	

OAT—Oral Anticoagulant Therapy, NOAC—Non vitamin K antagonist Anticoagulant, VKA—Vitamin K Antagonist.

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
