# Peer review of "Current Use of Oral Anticoagulation Therapy in Elderly Patients with Atrial Fibrillation: Results from an Italian Multicenter Prospective Study—The ISNEP Study"

_jpm, 2022, doi:10.3390/jpm12091419_

Round 1
Reviewer 1 Report
Dear Sir/Madam,
I had the opportunity to act as a reviewer on the recent submission by De Stefano et al. to the Journal of Personalized Medicine.
The authors present interesting research on the topic of real-world use of oral anticoagulation in a population of elderly patients in Italy.
The manuscript is well written and the results are interesting and of clinical interest.
However, following issues need to be addressed:
- I recommend converting the abstract to an abstract with a classic structure: background, aim, methods, results and conclusion. Also, in the abstract the age of the included patients is missing: “aged ≥ years”, line 28.
- I find figure 1 somewhat confusing: since the current guidelines do not use CHADS2 score anymore, I recommend eliminating it.
- On lines 259-260 the authors claim to have taken a snapshot of patient’s adherence. However, adherence has not been studied, although the authors mention under Conclusion, that “a higher level of satisfaction about the therapy, which, in perspective, can be associated with a more complete adherence throughout the course of it”. Regarding this matter, please remove the word “adherence” on line 260.
- I recommend a spell check performed by a native speaker (i.e., “can have been led” on line 217).
Best regards,
Author Response
We are thankful for the useful and precise suggestions, the file attached contains our corrections

Reviewer 2 Report
In the manuscript titled “Current use of oral anticoagulation therapy in elderly patients with atrial fibrillation: results from an Italian multicenter prospective study. The ISNEP Study”, the study showed the real-world use of OAT in a population of elderly patients in the Italian national territory, as well as the tolerance and patients satisfaction about the therapy. It plays a certain role in clinical guidelines. In this study, the number of people in the VKA group (N = 45) and NOAC (N = 596) is quite different, which may have a certain impact on the accuracy of the experimental conclusions. The sample size of the VKA group needs to be increased. In addition, there is much to improve in academic paper writing.
Author Response
We are thankful for the useful and precise suggestions, he attached file contains the response and the corrections.

Reviewer 3 Report
De Stefano et al. present the reuslts of a croos-sectional registry of elderly aptients with AF. The results are novel and the topic is interesting but there are some aspects to clarify:
1. I think that you did not perform a "prospective" registry. You collected the date restrospectively because all the information about embolisms are referred to the time before the inclusion.
2. Maybe you should nclude among the main results of your registry that patients under NOACs presented few embolic events during the therapy.
3. Please define properly the events that you reported (e.g. what definition of bleeding did you consider?) and how you collected them (medical interview? review of medical reports?).
4. In fact patients under NOACs did not present a significant reduction in bleeding. You should not mention it among the main findings of your registry.
5. You should consider performing a multivariate analysis with the main findings of your study.
6. I do not understand the answer of the questionnaire about quality of life. Patients under NOACs significantly presented more "complete" changes in their quality of life.
7. Minor corrections: please mention the cut-off point for age in the abstract, review the time under VKA therapy in table 2, do not repeat the same information in supplemantary material and in the main manuscript.
Author Response
We are thankful for the useful and precise suggestions, the attached file contains our response and corrections.

Round 2
Reviewer 1 Report
Dear Sir/Madam,
Thank you for reviewing the manuscript and addressing the mentioned issues.
Unfortunately, the issue regarding academic paper writing still remains: for instance, line 238 of the revised manuscript contains the following expression: “Both of these reasons could explicate”.
Therefore, I strongly recommend addressing this issue before publishing.
Best regards,
Author Response
Dear Reviewer, thank you for your reply.
The Manuscript has been analyzed and corrected by the English Editing service of MDPI site.
Best Regards
Reviewer 2 Report
The authors referred that “The aim of this study was not to compare head-to-head NOAC vs. VKA treatments, but to describe the prescription patterns of these drugs and the characteristics of AF patients receiving either NOACs or VKAs, for thromboembolic prevention, in a real-world elderly patient setting.” The article can provide a certain reference value for clinical work, but the meaning is also very limited.
The authors referred that “The study included 641 outpatients from 23 Italian participating centers, and only 45 (7%) were on treatment with VKA, which reflects the current use of OAT in Italy and in most Western countries. Due to the observational nature of this study, we cannot modify the sample of VKA subgroup as well as make direct comparisons between different OAT strategies.” If the author can't do it, the experimental results may not be so convincing.
Writing quality have be improved after revision. Moreover, it would be better if the authors could change the table format into a three-line table in an academic paper (Table 1-3).
Author Response
Dear Reviewer, we thank you for the time spent in reviewing and improving our Study. We concur that the Study has some limitations; however, the value of the Study lays in the description of the OAT prescription patterns in a real-world multicenter registry, involving 23 center throughout the italian national territory, and to the patients' satisfaction degree with NOACs treatment. Since our study was not designed to directly compare NOACs vs VKAs strategies, in our opinion the limited sample size is not a very strong limitation. This concept has further been emphasized in the limitations section.
The Manuscript has been analyzed and corrected by the English Editing service of MDPI site.
Best Regards